# Determination of CYP450 Expression Levels in the Human Small Intestine by Mass Spectrometry-Based Targeted Proteomics

**DOI:** 10.3390/ijms222312791

**Published:** 2021-11-26

**Authors:** Alexia Grangeon, Valérie Clermont, Azemi Barama, Fleur Gaudette, Jacques Turgeon, Veronique Michaud

**Affiliations:** 1CRCHUM, Centre de Recherche du Centre Hospitalier de l’Université de Montréal, 900 St. Denis Street, Montreal, QC H2X 0A9, Canada; alexia.grangeon.chum@ssss.gouv.qc.ca (A.G.); valerie.clermont.1@umontreal.ca (V.C.); fleur.gaudette.chum@ssss.gouv.qc.ca (F.G.); 2CHUM, Centre Hospitalier de l’Université de Montréal, 1000 St. Denis Street, Montreal, QC H2X 0C1, Canada; azemi.barama.chum@ssss.gouv.qc.ca; 3Faculty of Pharmacy, Université de Montréal, 2940 Chemin de la Polytechnique, Montreal, QC H3T 1J4, Canada; j.turgeon@umontreal.ca or; 4Precision Pharmacotherapy Research and Development Institute, Tabula Rasa HealthCare, 13485 Veterans Way, Orlando, FL 32827, USA

**Keywords:** protein absolute quantification, LC-MS/MS, targeted proteomics, CYP450, human intestine, interindividual variabilities

## Abstract

The human small intestine can be involved in the first-pass metabolism of drugs. Under this condition, members of the CYP450 superfamily are expected to contribute to drug presystemic biotransformation. The aim of this study was to quantify protein expression levels of 16 major CYP450 isoforms in tissue obtained from nine human organ donors in seven subsections of the small intestine, i.e., duodenum (one section, N = 7 tissue samples), jejunum (three subsections (proximal, mid and distal), N = 9 tissue samples) and ileum (three subsections, (proximal, mid and distal), N = 9 tissue samples), using liquid chromatography tandem mass spectrometry (LC-MS/MS) based targeted proteomics. CYP450 absolute protein expression levels were compared to mRNA levels and enzyme activities by using established probe drugs. Proteins corresponding to seven of sixteen potential CYP450 isoforms were detected and quantified in various sections of the small intestine: CYP2C9, CYP2C19, CYP2D6, CYP2J2, CYP3A4, CYP3A5 and CYP4F2. Wide inter-subject variability was observed, especially for CYP2D6. CYP2C9 (*p* = 0.004) and CYP2C19 (*p* = 0.005) expression levels decreased along the small intestine. From the duodenum to the ileum, CYP2J2 (*p* = 0.001) increased, and a trend was observed for CYP3A5 (*p* = 0.13). CYP3A4 expression was higher in the jejunum than in the ileum (*p* = 0.03), while CYP4F2 expression was lower in the duodenum compared to the jejunum and the ileum (*p* = 0.005). CYP450 protein levels were better correlated with specific isoform activities than with mRNA levels. This study provides new data on absolute CYP450 quantification in human small intestine that could improve physiologically based pharmacokinetic models. These data could better inform drug absorption profiles while considering the regional expression of CYP450 isoforms.

## 1. Introduction

The liver and small intestine are involved in drug biotransformation due to their high content of drug metabolizing enzymes [1,2,3,4,5]. Metabolizing enzymes found in both the liver and small intestine include the superfamily of cytochrome P450 (CYP450) monooxygenases, UDP-glucuronosyltransferases, sulfotransferases, flavin monooxygenases, carboxylesterases, N-acetyl transferases, glutathione S-transferases and alcohol dehydrogenases [2,3,4,5,6,7,8].

Most drugs used clinically are administered orally; however, oral drug bioavailability could be limited by chemical instability, poor solubility, poor absorption (poor passive or transporter-limited diffusion) and extensive pre-systemic metabolism [4,7,9,10,11,12,13,14,15]. The liver has higher expression levels of total CYP450s than the small intestine, but intestinal metabolism by some CYP450 isoforms, e.g., CYP3A4, could be responsible for the pre-systemic metabolism of some drugs [3,4,16]. Furthermore, the pattern of CYP450 isoform expression appears to vary from one section of the small intestine to the other [17,18]. Although some pieces of information are available, full characterization of CYP450-mediated metabolism in the human gut wall is not complete as data are limited to the proximal region of the small intestine. Information is also often based on mRNA levels or enzyme activity assays rather than direct protein quantification [3,12,17,19,20,21,22,23]. Consequently, absolute quantification of major CYP450 isoforms in the human small intestine is needed in order to better understand inter-subject and intrasubject variability in pre-systemic metabolism and drug pharmacokinetics [1,10,18,24,25,26,27].

In recent years, high performance liquid chromatography/tandem mass spectrometry (HPLC-MS/MS) methods for the quantification of proteins have emerged [18,24,25,26,27,28,29,30]. These methods are based on the quantification of proteotypic peptides obtained by tryptic digestion of CYP450s and the use of stable isotope-labeled peptides as internal standards. The literature contains few reports of CYP450 quantification by LC-MS/MS in the human small intestine [18,24,26,27]. In previous studies, important CYP450 were not quantified or studies were performed with a very limited number of tissue samples from a limited number of donors in a limited number of sections along the small intestine [18,24,26,27].

We have developed and validated an absolute quantification method by HPLC-MS/MS in order to determine expression levels of 14 CYP450 isoforms in various sections of the human small intestine [25]. The aim of the current study is to use this assay to determine protein expression level of these CYP450 isoforms (in addition to two others, i.e., CYP2A6 and CYP4A11) in tissues from several donors and in various sections along the small intestine. We also aimed to compare CYP450 protein expression to mRNA levels and enzymes activities using established probe drugs for CYP450 isoforms.

## 2. Results

### 2.1. CYP2A6 and CYP4A11 Quantification Method Validation

Quantifications of CYP2A6 and CYP4A11 were added to our previously published protein quantification method [25]. Detailed results and validation of the CYP2A6 and CYP4A11 isoforms methods are described in Text S1. In brief, two proteotypic peptides were selected for CYP2A6 (GTGGANIDPTFFLSR and DPSFFSNPQDFNPQHFLNEK) and CYP4A11 (VATALTLLR and NAFHQNDTIYSLTSAGR) (Appendix A). The assay was linear over a range of 0.1 to 15 nM. The LLOQ and intra/inter-day precisions were better than 19.7% and 11.4%, respectively; the LLOQ and intra/inter-day accuracies were 80.7–118.8% and 88.0–112.1%. According to current U.S. FDA guidelines for bioanalytical methods, our assay was also validated for selectivity, stability and the matrix effect (Appendix A) [31]. Representative chromatograms of the overlay of intestinal microsomes from one tissue donor—a blank with internal standard and a standard at 2 nM—are shown in Appendix A.

### 2.2. Absolute Intestinal Expression of CYP450 Isoforms

Seven of sixteen analyzed CYP450 isoforms were observed in the small intestine: CYP2C9, CYP2C19, CYP2D6, CYP2J2, CYP3A4, CYP3A5 and CYP4F2. CYP2C19, CYP2J2, CYP3A4, CYP3A5 and CYP4F2 were measured in all nine donors and in each section, whereas CYP2C9 and CYP2D6 expression were measured in 98.4% and 91.8% of the intestinal specimens, respectively (Appendix A).

Relative patterns of CYP450 expression for the duodenum, jejunum and ileum sections are presented in Figure 1. In all sections, CYP3A4 exhibited the highest level of protein expression, accounting for almost 78% of total CYP450 content in the small intestine.

A scatterplot of the seven CYP450 measured in the different sections of small intestine, i.e., duodenum (N = 7); proximal, mid and distal jejunum (N = 9); and proximal, mid and distal ileum (N = 9) for each donor is presented Figure 2.

CYP3A4 protein expression along the small intestine ranged from 34 ± 16 (distal ileum) to 52 ± 30 (proximal jejunum) pmol/mg protein (Table 1). When anatomical sections were considered, i.e., duodenum (seven samples), mean of jejunum (27 samples) and mean of ileum (27 samples), CYP3A4 expression level was greater in the jejunum vs. the ileum (*p* = 0.029; Table 2). The highest interindividual variability was observed in the distal jejunum section where CYP3A4 protein expression was 6.8 ± 3.3 to 98 ± 26 pmol/mg protein.

CYP4F2 (6.7–11.6%) and CYP2C9 (4.5–9.5%) were the second and third most expressed isoforms in the small intestine (Figure 1). CYP4F2 mean expression ranged from 3.6 ± 2.9 (duodenum) to 6.5 ± 4.1 (distal jejunum) pmol/mg protein along the small intestine (Table 1). CYP4F2 expression was higher in the jejunum and ileum compared to the duodenum when anatomical sections were considered (*p* = 0.005; Table 2). Interindividual variability observed was mainly attributed to two donors, i.e., IN003 and IN005, for which they showed higher expression levels than all other donors for all sections. CYP2C9 protein expression decreased significantly from an average of 6.9 ± 3.9 pmol/mg protein in the proximal part of the jejunum to 1.7 ± 0.74 pmol/mg protein in the distal part of the ileum (*p* = 0.002, Table 1). This pattern of expression was observed in all donors between the various subsections.

CYP2C19, CYP2J2 and CYP3A5 proteins were expressed at 1–5% of all CYP450s measured in human intestinal tissue (Figure 1). As observed with CYP2C9, CYP2C19 expression decreased significantly from 1.4 ± 0.83 pmol/mg protein (duodenum) to 0.32 ± 0.18 (distal ileum) (*p* = 0.011; Table 1). Moreover, CYP2C19 exhibited higher variability in the jejunum sections than ileum sections. The expression levels of 0.08 ± 0.07 to 3.0 ± 0.43 pmol/mg protein in the jejunum and 0.03 ± 0.03 to 0.81 ± 0.23 pmol/mg protein were observed in ileum sections. CYP2J2 and CYP3A5 interindividual variabilities were relatively small compared to other CYP450 isoforms. CYP2J2 expression increased significantly along the three parts of the small intestine when anatomical sections were considered (*p* = 0.001; Table 2). CYP3A5 expression in the small intestine varied from 0.52 ± 0.21 (duodenum) to 0.83 ± 0.24 (mid-ileum) pmol/mg protein (Table 1). When anatomical sections were considered, CYP3A5 expression tended to increase along the small intestine, but it did not reach statistical significance (*p* = 0.132; Table 2).

Finally, CYP2D6 was expressed at low levels (<1%) while CYP1A1, CYP1A2, CYP1B1, CYP2A6, CYP2B6, CYP2C8, CYP2E1, CYP3A7 and CYP4A11 were undetectable in all donors and in all sections analyzed. CYP2D6 expression appeared constant along all intestinal sections and varied from 0.38 ± 0.41 (duodenum) to 0.76 ± 0.69 (distal ileum) pmol/mg protein (Table 1). CYP2D6 showed a high intra-section variability in all sections as some donors had very low CYP2D6 expression, e.g., IN001, IN004 and IN009.

### 2.3. Correlation between Protein Expression, mRNA Levels and Enzyme Activity

mRNA levels and enzyme activities were evaluated in a previous paper [17]. In brief, 17 CYP450 were analyzed for mRNA expressions (seven sections/subject); of the 17 CYP450, 14 were detected in donor tissues. mRNA levels of CYP3A4, CYP2C9, CYP2C19 and CYP2J2 were the highest (>5%). CYP4F2, CYP4F12 and CYP3A5 displayed intermediate level (1–5%), whereas CYP1A1, CYP2B6, CYP4A11, CYP1B1, CYP2D6, CYP2E1 and CYP2C8 were measured at low levels (<1%). The mRNAs of these CYP450 were measured in most sections and in all patients (>97%). CYP1A1 and CYP2C8 mRNAs were measured in ≈70% of the intestinal specimens. CYP1A2, CYP2A6 and CYP3A7 mRNA were not detected in any patients. Correlation between expression levels of mRNA and protein were evaluated in nine patients within seven sections, i.e., duodenum, jejunum (proximal, mid and distal) and ileum (proximal, mid and distal) for the seven CYP450 protein levels determined by LC-MS/MS. Protein and mRNA expressions were well correlated for CYP2D6 (r_s_ = 0.63); moderately correlated for CYP2C9, 2C19, 2J2 and 3A4 (r_s_ = 0.41–0.53); and poorly correlated for CYP4F2 (r_s_ = 0.26). CYP3A5 protein and mRNA expressions showed a negative and poor correlation (Table 3).

The functional activity of nine CYP450 isoforms was measured in microsomes prepared from five sections of the small intestine, i.e., duodenum, jejunum (proximal and mid) and ileum (proximal and mid). Overall CYP450 activities ranged as CYP3A4 > CYP2J2 > CYP2E1 > CYP2C9 > CYP4A11 > CYP2C19 > CYP2B6 > CYP2D6 > CYP2C8. CYP2C9, CYP2B6, CYP2J2, CYP2E1 and CYP3A4 activities were measured in more than 84% of the intestinal specimens. Only 40 to 63% of specimens showed CYP2B6 and CYP2D6 activities. Protein expression and enzyme activities correlation were evaluated within these five sections for six CYP450 measured by LC-MS/MS. Correlation for CYP4F2 was not assessed as CYP4F2 activity was not observed. High correlations for CYP2C9, 2C19, 2D6 and 3A4 (r_s_ = 0.68–0.82) were observed while CYP2J2 (r_s_ = 0.44) showed a moderate correlation (Table 3).

Altogether, for all CYP450s, protein expression correlated better with enzymes activities than mRNA expression. Correlation analyses obtained with Spearman’s correlation analyses are reported in Table 3.

### 2.4. Effect of Genotype, Sex, Age and Body-Mass Index (BMI) on CYP450 Protein Expression

Three subjects were homozygous for the *CYP2C9*1* allele, while five subjects were heterozygous for allelic variants; one subject was homozygous for allelic variants. These allelic variants (**2* or **3*) are associated with decreased function or non-functioning allele. Three subjects were found to have the heterozygous *CYP2C19*1/*17* allelic variant (associated with an ultra-rapid metabolizer phenotype), and six subjects were homozygous *CYP2C19*1/*1*. No donors were assigned a *CYP2C19* allelic variants (linked to poor metabolizer phenotype). Six donors were assigned a CYP2D6 extensive metabolizer phenotype and three an intermediate metabolizer phenotype; no donor carried the *CYP2D6* duplication or multiplication gene. Eight subjects carried the wild type allele for *CYP3A4*, and one donor was heterozygous for *CYP3A4*22* (related to a proposed decreased enzymatic function). No subject carried the wild type allele for *CYP3A5*, as all donors were homozygous for the *CYP3A5*3* allele. These genotypes showed no correlation with CYP450 protein expression. The wild type *CYP2J2*1* allele was carried in all donors except for one donor who was heterozygous for the *CYP2J2*7* allelic variant, which has been associated with decreased enzyme activity. However, the only donor carrying this variant seemed to have higher CYP2J2 protein expression [32]. CYP450 allele frequencies are reported in Appendix A.

No CYP450 was correlated with sex. We found that CYP2C19 and CYP2J2 protein expression tended to be higher in donors under 50 years of age (*p* = 0.0635). CYP2J2 appeared to be slightly higher in individuals with a BMI < 30 (*p* = 0.0635). Correlation analyses are preliminary due to the limited number of donors in each group (Appendix A).

## 3. Discussion

In this study, we aimed to determine the protein expression of 16 CYP450 isoforms in a representative sample size of donors and in various subsections of the human small intestine. Of these sixteen isoenzymes examined, seven were detected and quantified in all donors. Their relative expression levels are ranked as follows: CYP3A4 (77–78%) > CYP4F2 (7–12%) ≅ CYP2C9 (5–10%) > CYP2J2 (1–2.6%) ≅ CYP2C19 (0.8–2.6%) ≅ CYP3A5 (1–1.6%) > CYP2D6 (0.7–1%) (Figure 2). CYP1A1, CYP1A2, CYP1B1, CYP2A6, CYP2B6, CYP2C8, CYP2E1, CYP3A7 and CYP4A11 were undetectable in all subsections of all donors. As our tissue samples were from nine donors, the classification of CYP450 expression level in a larger cohort could differ for CYP450 with low expression (≤5%). Our results demonstrated highest correlations between protein expression and enzyme activity than between protein expression and mRNA for CYP2C9, CYP2C19, CYP2D6, CYP2J2, CYP3A4 and CYP4F2.

It has long been recognized that CYP450s are expressed in the small intestinal mucosa. Early studies have shown that 50 to 70% of spectrally determined CYP450 content was related to the CYP3A subfamily [19,33]. CYP3As appeared localized in enterocytes (epithelial cells), and microsomal content and activity were the highest in the proximal region [19,34,35]. Using polyclonal antibodies, the CYP2C subfamily was reported as the second most abundant CYP450s, while studies by Paine et al. established the relative contribution of other isoforms (CYP3A, CYP2C9, CYP2C19, CYP2D6 and CYP2J2) [3,36,37,38].

To date, three groups have determined CYP450 protein expression in the human small intestine by LC-MS/MS [18,24,26]. In agreement with previous reports (CYP3A4, 22–33 pmol/mg; CYP2C9, 1.4–3.2 pmol/mg), our results confirmed that CYP3A4 (34–52 pmol/mg) was the most abundant CYP450 isoform, followed by CYP2C9 (1.7–6.9 pmol/mg) [18,24,26]. Expression levels for CYP2C19, CYP2D6, CYP2J2 and CYP3A5 were also in the same order of magnitude than levels reported in other studies [18,24,26].

Interestingly, CYP4F2 (3.6–6.5 pmol/mg)—which was not quantified in any other study—was also highly expressed in all subsections of the small intestine in all donors. Michaels et al. previously showed high expression of CYP4F2 in the liver [5]. More studies are needed to understand CYP4F2 role in the metabolism of endogenous compounds and xenobiotics [5,39,40].

As mentioned, CYP1A1, CYP1A2, CYP1B1, CYP2A6, CYP2B6, CYP2C8, CYP2E1, CYP3A7 and CYP4A11 levels were not detected in the intestinal tissues analyzed in this study. Miyauchi et al. reported low levels of CYP1A1, CYP1A2, CYP1B1, CYP3A7 and CYP4A11 in intestinal tissues in less than half of their subjects [24]. Their results could be attributed to the use of a single peptide to quantify CYP450, lacking confirmation from a second peptide. Drodzik et al. did not detect CYP1A2, CYP2B6, CYP2C8 or CYP2E1 in any of their patients [18].

We observed that CYP2C9 and CYP2C19 levels decreased as we moved from the duodenum to the distal subsections of the ileum. Drozdzik et al. also described decreased expression of CYP2C9 and CYP2C19 from the duodenum to the ileum [18]. We also observed that CYP2J2 levels increased as we moved towards the distal part of the ileum. In contrast to our observations and those of Drozdzik et al., Couto et al. did not observe a significant change in CYP2C19, CYP2C9 and CYP2J2 expression between subsections [26]. This could be due to the methodology used, as tissues were obtained from different donors in each part of the intestine (jejunum (N = 4) and ileum (N = 12)). CYP3A4 and CYP4F2 expressions were higher in the jejunum when anatomical sections were considered, i.e., duodenum (seven samples), mean of jejunum (27 samples) and mean of ileum (27 samples). These findings are in accordance with Paine et al. who measured higher levels of CYP3A4 in middle jejunum [19].

High inter-subject variability in CYP450 protein expression was observed in our study, especially for CYP2D6 (90-fold). Other CYP450 isoforms were somewhat less variable as expression varied 33-fold for CYP2C9, 14-fold for CYP3A4 and CYP2C19, 9-fold for CYP4F2 and CYP3A5, 6-fold for CYP2J2 and 3-fold for CYP3A5. Similar inter-subject variability was observed by other investigators [24]. Inter-subject variability could be due to the effect of medications taken by patients. Unfortunately, we did not have access to this information to conclude whether medications taken by donors could have induced the expression of CYP450s.

We observed a significant correlation between CYP450 protein expression levels and both mRNA levels and activities for CYP2C9, CYP2C19, CYP2D6, CYP2J2, CYP3A4 and CYP4F2. The observed correlation was stronger between protein levels and activities than between protein expression and mRNA levels. In a previous study from our group, Clermont et al. showed some degree of correlation between mRNA levels and enzyme activities for the same CYP450s. These results support the concept that CYP450 protein expression measured by LC-MS/MS is more representative of CYP450 activities than mRNA expression. Similar results were observed in the liver by Ohtsyki et al. and by others, although most studies reporting on the distribution of CYP450 in the human intestine are based on CYP450 mRNA levels [3,12,17,19,20,21,22,23,41].

## 4. Materials and Methods

### 4.1. Chemicals and Reagents

HPLC-MS grade water (H_2_O) was purchased from EMD Millipore (Billerica, MA, USA). HPLC-MS grade acetonitrile (ACN), MS grade trypsin protease, dithiothreitol (DTT), phosphate potassium, ammonium bicarbonate (NH_4_HCO_3_) and TRIS were obtained from Thermo Fischer Scientific (Waltham, MA, USA). Formic acid (FA), iodoacetamide (IAA), trifluoroacetic acid (TFA), bovine serum albumin (BSA), leupeptin, aprotinin, glycerol and phenylmethylsulfonyl fluoride (PMSF) were purchased from Sigma Aldrich (St. Louis, MO, USA). Proteotypic peptides and their stable isotope-labelled internal standards were synthetized by New England Peptides (Boston, MA, USA). All peptide purity was superior to 95%, and the concentration/net peptide content was determined by amino acid analysis.

### 4.2. Subjects

Study protocol was approved by the Centre Hospitalier de l’Université de Montréal (CHUM) institutional review board (CHUM-IRB; project 14.109). Human intestinal specimens were obtained in collaboration with Transplant Québec, a nonprofit organization that coordinates the organ donation process. All donors consented to organ donation for transplantation and research purposes. Small intestines from 9 donors were collected: mean age was 54 years (range 27–75), mean BMI was 26.9 (range 20.5–32.7), 56% were female and 67% were known smokers. Additional demographic characteristics are shown in Table 4.

### 4.3. Human Intestinal Tissue Preparation

The entire small intestine was obtained from 9 donors (except for the duodenum of 2 donors, as their pancreases were collected for transplantation). Patients identified for tissue donation were artificially kept alive to allow collection of multiple organs used for transplantation (removed first). Other organs were then made available for research purposes. The entire small intestine was collected on site at the donor hospital and processed within 3 to 6 h. Tissues were immediately immersed in a preservation cold Krebs-Ringer solution by surgeons and stored in an ice box for transportation to the research center. The three different regions of the small intestine (duodenum, jejunum and ileum) were visually identified and sectioned in pieces of 10 cm each. The obtained materials were washed twice in an ice-cold solution of Tris-KCl-EDTA buffer (100 mM, pH 7.4) and frozen in liquid nitrogen to be stored at −80 °C.

### 4.4. Intestinal Microsomes Preparation

CYP450 quantification using intestinal microsomes was performed for 6 or 7 different subsections (duodenum; proximal, mid and distal jejunum; and proximal, mid and distal ileum) along the small intestine for each donor, depending on the availability of the duodenum. For enzyme activity determination, intestinal microsomes were available from 5 subsections (duodenum, proximal-jejunum, mid-jejunum, proximal-ileum and mid-ileum) due to the limited amount of tissue available. The microsomal fraction from intestinal tissues was isolated by using a previously published protocol [42]. In brief, frozen tissues (8–13 g) from subsections were slightly thawed on a cold metal plate before mucosal scraping with disposable scalpels. The mucosal layer was immersed in an ice-cold solution of Tris-KCl-EDTA buffer (50 mM–150 mM–1 mM) containing dithiothreitol (0.5 mM), PMSF (0.01 mM), leupeptin (5 µg/mL) and aprotinin (50 µg/mL) and was homogenized on ice with a tissue homogenizer. The microsomal fraction was isolated by using a differential ultracentrifugation technique. The first centrifugation step was performed for 20 min at 10,000× *g* at 4 °C. The supernatant was transferred to a clean centrifuge tube and centrifuged for 90 min at 100,000× *g* at 4 °C. The resulting pellet was resuspended in Tris-KCl-EDTA buffer and centrifuged for 90 min at 100,000× *g* at 4 °C. The microsomal pellet was resuspended in a Tris-KCl-EDTA buffer with 10% glycerol. Intestinal microsomes were stored at −80 °C until further analyzed.

### 4.5. LC-MS/MS-Based Protein Quantification

An absolute quantification (AQUA) method was used to quantify various CYP450 isoforms in small intestine microsomes. This quantitative proteomic technique is based on the quantification of proteotypic peptides obtained by tryptic digestion of proteins and stable isotope-labeled peptides used as internal standards. As previously described, proteotypic peptides were selected by combining in silico and in vitro methods [25]. In silico methods were performed with UniProtKB/Swiss-Prot database and the mMass software (Prague, Czech Republic) in order to identify theoretical peptides and to eliminate peptides with unsuitable features (non-synonymous genetic polymorphisms, oxidative instability, missed cleavage, nonspecific peptide, etc.) [43]. In vitro methods were performed with recombinant CYP450s to exclude peptides if no saturation in the peptide amount produced was observed after a 16 h tryptic digestion. Two proteotypic peptides were selected for each CYP450 isoform.

Protein quantification was performed by a Thermo Scientific TSQ Quantiva triple quadrupole mass spectrometer (San Jose, CA, USA) coupled with the Thermo Scientific Ultimate 3000 XRS UHPLC system (San Jose, CA, USA) using pneumatic assisted heated electrospray ion source (HESI) [25]. Briefly, proteins were (1) diluted in NH_4_HCO_3_ (50 mM, pH 7.8); (2) reduced with dithiothreitol for 20 min at 60 °C; and (3) alkylated with iodoacetamide for 15 min at 37 °C. Tryptic digestion was achieved by incubating microsomes with trypsin for 16 h at 37 °C. Digestion was ended by the addition of 20 µL of an acidic solution containing the internal standards, and samples were centrifuged at 16,000× *g* for 10 min at 4 °C. The supernatant evaporated to dryness, and the dried extract was re-suspended with 30 µL of ACN:H_2_O:TFA (10:90:0.1, *v/v*). Chromatographic separation was performed using a Biobasic-8 100 × 1 mm, 5 µm analytical column (Thermo Scientific, Waltham, MA, USA) under a gradient program at a flow rate of 75 μL/min. The initial mobile phase (5:95, 0.1% FA in ACN: 0.1% FA in H_2_O, *v/v*) was maintained for 1 min. From 1–61 min, a linear gradient was applied up to a ratio of 32:68 and was reverted to initial conditions at 61 min for column re-equilibration. The column temperature was set at 40 °C, and the injection volume was 2 μL. Two or three SRM transitions were selected for each peptide, and the protein amount for each CYP450 isoform was calculated as the average of the two peptide quantification results. (Appendix A).

### 4.6. CYP2A6 and CYP4A11 Quantification

Quantifications of CYP2A6 and CYP4A11 were added to our previously published protein quantification method [25]. Standard (25 µM) and internal standard (5 µM) stock solutions were prepared in 0.1% TFA in water. Solutions of peptides and internal standards were infused into the mass spectrometer to obtain ionization conditions and MS/MS parameters. The SRM transitions selected for each peptide were based on the highest intensity ion for the precursor ion and lowest background. The calibration standards and quality control samples were prepared by fortifying 0.25 mg/mL of digested BSA with working solutions to generate an analytical range of 0.1 to 15 nM and were analyzed according to our published quantification method [25]. The SRM transitions and collision energy (Elab) are reported in Appendix A. CYPA6 and CYP4A11 quantification methods were applied to two commercially available pooled HLM, one commercially available pooled HIM and three individual in-house HIM preparations (Appendix A).

### 4.7. CYP450 mRNA Expression, CYP450 Activities and Genotypes

The determination of CYP450 mRNA levels, CYP450 isoform activities and genotypes was performed as described previously [17]. Briefly, total RNA was extracted from all available tissue subsections of patients’ small intestine, and relative mRNA expression was evaluated by quantitative real-time polymerase chain reaction (RT-qPCR). mRNA expression was determined in seven sections along the small intestine: duodenum, jejunum (proximal, mid and distal) and ileum (proximal, mid and distal). In vitro incubations were performed with nine CYP450 probe substrates (bupropion, CYP2B6; repaglinide, CYP2C8; tolbutamide, CYP2C9; S-mephenytoin, CYP2C19; bufuralol, CYP2D6; chlorzoxazone, CYP2E1; ebastine, CYP2J2; midazolam, CYP3A4/3A5; and lauric acid, CYP4A11) and analyzed by LC-MS/MS in order to evaluate enzymes activities. Incubations were performed in five different sections (duodenum, proximal jejunum, mid-jejunum, proximal ileum and mid-ileum) at substrate saturation. TaqMan RT-qPCR SNP Genotyping Assay (Applied Biosystems, Foster, CA, USA) was used to genotype the common variants for the 16 CYP450 isoforms quantified in this study (Appendix A).

### 4.8. Data and Statistical Analysis

CYP450 protein levels for each isoform were measured in duplicate in three independent experiments. For each CYP450 isoform, protein quantification was measured with two proteotypic peptides. The final CYP450 protein amount was calculated by using the average of the two protein quantification results obtained with two peptides. Statistical analyses were performed on GraphPad Prism 8.0 (GraphPad Software, La Jolla, CA, USA). Data were expressed using mean ± sd in order to summarize the characteristics of subjects. Continuous values determined at different intestine sections were analyzed using a linear mixed model with one fixed factor. The dependence among repeated measurements where modeled using a heterogeneous covariance matrix structure of correlations. As data are correlated, a Cholesky factorization was performed on the error distribution from the statistical model to verify the normality assumption with the Shapiro–Wilk tests. Brown and Forsythe’s variation of Levene’s test statistic was used to verify the homogeneity of variances. Posteriori comparisons were performed by using Tukey’s technique. The results were considered significant with *p*-values ≤ 0.05. All analyses were conducted using the statistical package SAS v9.4 (SAS Institute Inc., Cary, NC, USA). Spearman’s correlation analysis was used to evaluate the association between CYP450 protein amount and CYP450 activities or mRNA levels.

## 5. Conclusions

This study brings both new and additional supporting data on CYP450 absolute quantification in the human small intestine. Of the sixteen major CYP450 analyzed by LC-MS/MS, seven CYP450 were measured in various subsections of the small intestine. High interindividual variability was observed in the expression level of these isoforms, which correlated well with enzyme activity using specific probe drugs. We also observed that CYP2C9 and CYP2C19 expression levels decreased along the small intestine as we moved from the duodenum to the distal part of the ileum, while in contrast CYP2J2 levels increased. Data could contribute to the improvement of physiologically based pharmacokinetics (PBPK) models and influence the development of new pharmaceutical formulations to improve drug bioavailability while considering the regional expression of CYP450 isoforms. These data provide a better understanding of the role of small intestinal sections in the presystemic metabolism of drugs and help predict and understand the mechanisms of drug interactions.

## Figures and Tables

**Figure 1 ijms-22-12791-f001:**
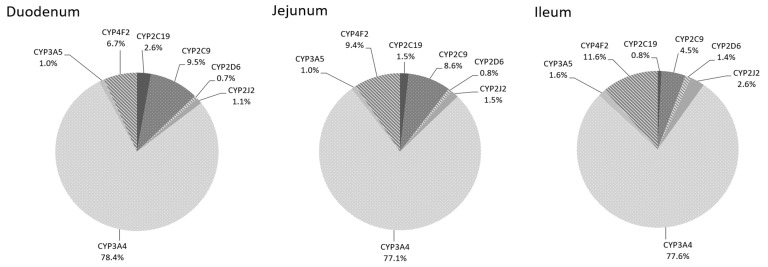
The intestinal pie chart of CYP450 from 9 donors in each intestinal section (duodenum, jejunum and ileum).

**Figure 2 ijms-22-12791-f002:**
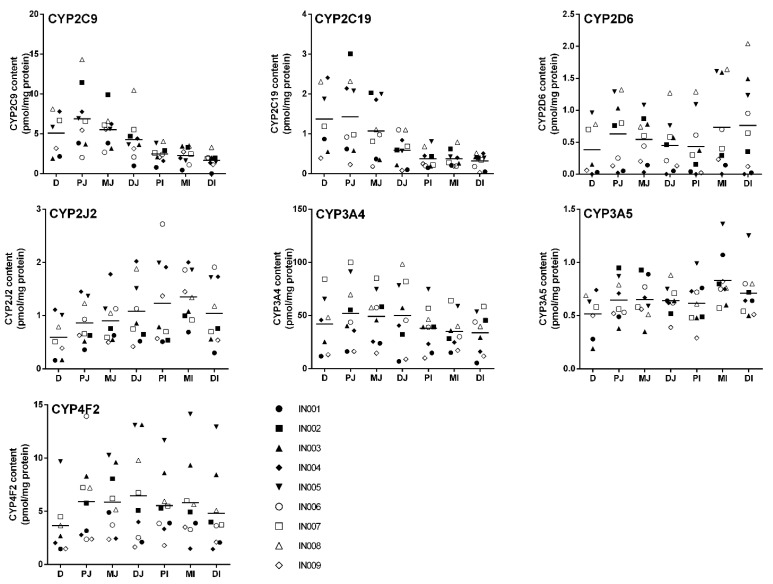
A scatterplot of the 7 CYP450 measured in different sections of the small intestine. Population means are indicated with lines, and each data point represents the mean of three independent digestions performed in duplicate. D, duodenum; PJ, proximal jejunum; MJ, mid-jejunum; DJ, distal jejunum; MI, mid-ileum; DI, distal ileum.

**Table 1 ijms-22-12791-t001:** Protein expression levels of 16 CYP450 isoenzymes in 7 intestinal subsections. Protein amounts are expressed in pmol/mg protein.

Protein	Duodenum				Proximal Jejunum				Medial Jejunum				Distal Jejunum				Proximal Ileum				Medial Ileum				Distal Ileum				*p* Value
	Mean	SD	Min–Max	N	Mean	SD	Min–Max	N	Mean	SD	Min–Max	N	Mean	SD	Min Max	N	Mean	SD	Min Max	N	Mean	SD	Min Max	N	Mean	SD	Min Max	N	
CYP1A1	ND				ND				ND				ND				ND				ND				ND				
CYP1A2	ND				ND				ND				ND				ND				ND				ND				
CYP1B1	ND				ND				ND				ND				ND				ND				ND				
CYP2A6	ND				ND				ND				ND				ND				ND				ND				
CYP2B6	ND				ND				ND				ND				ND				ND				ND				
CYP2C19	1.4 ^a,b,c,d^	0.83	0.39–2.4	7	1.4 ^e,f,g,h^	0.97	0.23–3.0	9	1.1	0.73	0.18–2.0	9	0.59 ^a,e^	0.37	0.08–1.1	9	0.38 ^b,f^	0.24	0.15–0.81	9	0.38 ^c,g^	0.19	0.19–0.79	9	0.32 ^d,h^	0.18	0.03–0.52	9	0.011
CYP2C8	ND				ND				ND				ND				ND				ND				ND				
CYP2C9	5.1 ^a^	2.6	1.9–8.1	7	6.9 ^b,c,d^	3.9	2.0–14	9	5.5 ^e,f,g^	2.2	2.7–9.9	9	4.3	3.2	0.97–10	9	2.5 ^b,e^	1.0	0.78–4.0	9	2.3 ^c,f^	1.1	0.44–3.5	9	1.7 ^a,d,g^	0.74	0–3.3	8	0.002
CYP2D6	0.38	0.41	0–0.96	7	0.62	0.53	0.02–1.3	9	0.54	0.36	0.03–1.1	9	0.45	0.39	0–1.3	9	0.43	0.48	0–1.3	8	0.73	0.69	0–1.6	8	0.76	0.69	0–2.0	8	0.193
CYP2E1	ND				ND				ND				ND				ND				ND				ND				
CYP2J2	0.59 ^a,b,c,d,e,f^	0.38	0.16–1.1	7	0.86 ^a,g^	0.40	0.36–1.5	9	0.90 ^b,h^	0.41	0.50–1.8	9	1.1 ^c^	0.50	0.42–2.0	8	1.2 ^d^	0.80	0.50–2.7	9	1.4 ^e,g,h^	0.47	0.69–2.0	9	1.0 ^f^	0.57	0.30–1.9	9	<0.001
CYP3A4	42	27	12–84	7	52	30	16–100	9	49 ^a^	24	15–85	9	50	27	6.8–98	9	38	20	9.9–75	9	35	17	15–64	9	34 ^a^	16	5.3–58	9	0.035
CYP3A5	0.52	0.21	0.19–0.74	7	0.64	0.19	0.38–0.95	9	0.65	0.19	0.35–0.93	9	0.64	0.13	0.39–0.88	9	0.62	0.21	0.29–0.99	9	0.83	0.24	0.57–1.4	9	0.71	0.27	0.50–1.3	9	0.13
CYP3A7	ND				ND				ND				ND				ND				ND				ND				
CYP4A11	ND				ND				ND				ND				ND				ND				ND				
CYP4F2	3.6 ^a,b,c,d^	2.9	1.5–9.7	7	5.9 ^a^	3.8	2.4–14	9	5.9 ^b^	2.9	2.4–10	9	6.5	4.1	1.6–13	9	5.5	3.0	1.8–12	9	5.8 ^c^	3.8	1.5–14	9	4.8 ^d^	3.9	1.5–13	9	0.005

^a,b,c,d,e,f,g,h^*p* value were < 0.05 between the two mentioned sections (per isoenzyme) CYP2D6 and CYP2C9 were not detected in all patients. ND; not detectable.

**Table 2 ijms-22-12791-t002:** Protein expression levels of 16 CYP450 isoenzymes in 3 intestinal subsections. Protein amounts are expressed are expression in pmol/mg prot.

Protein	Duodenum			Jejunum			Ileum			*p* Value
	Mean	SD	Min–Max	Mean	SD	Min–Max	Mean	SD	Min–Max	
CYP1A1	ND			ND			ND			
CYP1A2	ND			ND			ND			
CYP1B1	ND			ND			ND			
CYP2A6	ND			ND			ND			
CYP2B6	ND			ND			ND			
CYP2C19	1.4 ^a^	0.83	0.39–2.4	1.0 ^b^	0.63	0.08–3.0	0.36 ^a,b^	0.19	0.05–0.18	0.005
CYP2C8	ND			ND			ND			
CYP2C9	5.1 ^a^	2.6	1.9–8.1	5.6 ^b^	2.7	0.97–14	2.1 ^a,b^	0.87	0–4.0	0.004
CYP2D6	0.38	0.41	0–0.96	0.54	0.41	0–1.3	0.64	0.61	0–2.0	0.203
CYP2E1	ND			ND			ND			
CYP2J2	0.59 ^a,b^	0.38	0.16–1.1	0.95 ^a^	0.45	0.36–2.0	1.2 ^b^	0.61	0.30–2.7	0.001
CYP3A4	42	27	12–84	50 ^a^	27	6.8–100	36 ^a^	18	5.3–75	0.029
CYP3A5	0.52	0.21	0.19–0.74	0.65	0.11	0.35–0.95	0.72	0.21	0.29–1.4	0.132
CYP3A7	ND			ND			ND			
CYP4A11	ND			ND			ND			
CYP4F2	3.6 ^a,b^	2.9	1.5–9.7	6.1 ^a^	3.6	1.7–14	5.4 ^b^	3.5	1.5–14	0.005

^a,b^*p* values were < 0.05 between the two mentioned sections (per isoenzyme). CYP2D6 and CYP2C9 were not detected in all patients. ND; not detectable.

**Table 3 ijms-22-12791-t003:** Correlation analysis between mRNA levels, activity levels and protein content with the Spearman’s rank test (coefficient r_s_, n = 9).

Protein	mRNA vs. Protein	Activity vs. Protein
CYP1A1	-	-
CYP1A2	-	-
CYP1B1	-	-
CYP2A6	-	-
CYP2B6	-	-
CYP2C8	-	-
CYP2C9	0.5307 *	0.8179 *
CYP2C19	0.4940 *	0.7278 *
CYP2D6	0.6342 *	0.6750 *
CYP2E1	-	-
CYP2J2	0.4130 *	0.4378 *
CYP3A4	0.5066 *	0.6918 *
CYP3A5	−0.0664	-
CYP3A7	-	-
CYP4A11	-	-
CYP4F2	0.2559 *	-

* *p* value ≤ 0.05.

**Table 4 ijms-22-12791-t004:** Donors of small intestinal tissue demographic characteristics.

Subject	Sex	Age (Years)	BMI (kg/m^2^)	Cause of Death	Smoker	Comorbidities
IN001	M	73	31.3	Cerebrovascular accident	No	Hypertension, hyperlipidemia, diabetes, coronaropathy
IN002	F	49	32.7	Cerebrovascular accident	Yes	Multiple sclerosis, hypertension
IN003	F	75	24.0	Cerebrovascular accident	Yes	Multiple sclerosis, glaucoma
IN004	M	42	25.9	Anoxia	Yes	None
IN005	M	52	23.9	Head trauma	Yes	Hypertension
IN006	M	27	20.5	Anoxia	Yes	None
IN007	F	58	30.1	Cerebrovascular accident	Yes	Hyperthyroidism
IN008	F	49	23.2	Anoxia	NA	None
IN009	F	65	30.4	Anoxia	No	Hypertension, diabetes, cholangiocarcinoma

BMI, Body Mass Index; F, female; M, Male; NA, not available.

## Data Availability

All data are provided in the article and the Appendix A.

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
