# Peer review of "Determination of CYP450 Expression Levels in the Human Small Intestine by Mass Spectrometry-Based Targeted Proteomics"

_ijms, 2021, doi:10.3390/ijms222312791_

Round 1
Reviewer 1 Report
The paper describes targeted MS quantitation of 16 CYP450 isoforms along human small intestines of 9 donors (7 isoforms detected under LOD of the method). Data collected for the proteins are supplemented by previously reported mRNA values and activities for the isoforms, which makes the dataset unique and valuable for the field. I, however, recommend revision in data processing and interpretation without additional experiments.
Major points
- The previous art. CYP450s are in focus of extensive studies since 1970s. In this context, some efforts, I believe, were made to estimate their levels by antibodies. Please find those data for some of them (at least, in recent big data project called The Protein Atlas https://www.proteinatlas.org/) and provide at least rough comparisons with your results on protein levels.
- Comparisons between intestine sections. In your data, you deal with typical longitudinal rows. Thus, their comparison as independent data sets is not quite correct, as they are connected. Statistical trends between the intestine locations must be established by regression analysis e.g. using Mixed Linear Models, instead of Kruskal-Wallis. Please reanalyse your results. The same consideration relates to the correlation analysis, as each subject (donor) must be considered longitudinally (in spite gaps in duodenal data).
- Variance and CYP450 induction. Some of enzymes under study are induced by xenobiotics. This fact may explain the variance of them, which was not done in the paper. Ideally, information of drug use in donors would be helpful, but this latter may not by available. Any way, please consider the variance in light of induction.
Minor points
- Abstract is difficult for perception as it contains lots of brackets and excessive p-values reported. Please simplify. I think the abstract will be modified upon statistical reprocessing data.
- I recommend a submission of the SRM data to Proteomexchange or other data repository in accordance to the concept of open data shared by the Publisher.
Reviewer 2 Report
General comment:
This manuscript, entitled “Determination of CYP450 expression levels in the human small intestine by mass spectrometry-based targeted proteomics”, authored by Grangeon et al., reports a spectroscopy correlation between p450 expression. The used sensitive LCMS technique in profiling p450. This approach is suitable for understanding the overall expression and distribution in a tissue/organ-specific manner. In my opinion, this is a valuable work and is suitable for publication in the International Journal of Molecular Sciences after the authors have addressed the following comments and questions:
Major questions:
- What is the broader impact of this work? – conclusion should be more precise.
- The author should be careful in claiming new informative data as this work was reported previously, although the sample size was small. Still, in this work sample size is not great. The overall domination of cyp3A4 can be expected for any sample size, but trace expression of other p450 certainly will change if the sample size is more.
- Few P450s though expressed in good quantity, but didn’t show any activity – what are the possible reason the author expected that P450?
- Is there any physiological correlation author can suggest here for decreasing expression of certain p450 from duodenum to ileum?
Round 2
Reviewer 1 Report
No further comments. I can only wish a good luck to authors.